# A reproducibility study of "User-item fairness tradeoffs in recommendations"

**Sander Honig**                                  *sander.honig@student.uva.nl*
*University of Amsterdam*

**Elyanne Oey**                                   *elyanne.oey@student.uva.nl*
*University of Amsterdam*

**Lisanne Wallaard**                              *lisanne.wallaard@student.uva.nl*
*University of Amsterdam*

**Sharanda Suttorp**                              *sharanda.suttorp@student.uva.nl*
*University of Amsterdam*

**Clara Rus**                                     *c.a.rus@uva.nl*
*University of Amsterdam*

**Reviewed on OpenReview:** *https://openreview.net/forum?id=vltzxxhzLU*

## Abstract

Recommendation systems are necessary to filter the abundance of information presented in our everyday lives. A recommendation system could exclusively recommend items that users prefer the most, potentially resulting in certain items never getting recommended. Conversely, an exclusive focus on including all items could hurt overall recommendation quality. This gives rise to the challenge of balancing user and item fairness. The paper "User-item fairness tradeoffs in recommendations" by Greenwood et al. (2024) explores this tradeoff by developing a theoretical framework that optimizes for user-item fairness constraints. Their theoretical framework suggests that the cost of item fairness is low when users have varying preferences compared to each other, and may be high for users whose preferences are misestimated. They empirically measured these phenomena by creating their own recommendation system on arXiv preprints, and confirmed that the cost of item fairness is low when users have preferences that differ from one another. However, contrary to their theoretical expectations, misestimated users do not encounter a higher cost of item fairness. This study investigates the reproducibility of their research by replicating the empirical study. Additionally, we extend their research in two ways: (i) verifying the generalizability of their findings on a different dataset (Amazon books reviews), and (ii) analyzing the tradeoffs when recommending multiple items to a user instead of a single item. Our results further validate the claims made in the original paper. We concluded the claims hold true when recommending multiple items, with the cost of item fairness decreasing as more items are recommended.

## 1 Introduction

Recommendation systems have become increasingly popular due to the rapid expansion of the internet and the vast amount of information it holds (Lü et al. (2012)). They are necessary to filter the abundance of information, ensuring users are just shown a small selection of relevant items rather than being overwhelmed by information overload. One approach is to exclusively recommend items that the user prefers the most, however, this could result in certain items never getting recommended. To prevent this, algorithmic techniques have been developed to improve item fairness in recommendations (Mehrotra et al. (2018); Wang et al.

(2021)). However, too much focus on item fairness may lead to users getting less relevant recommendations, thereby compromising user fairness and, in turn, hurting the overall recommendation quality Wang et al. (2023). This shows the intricacy in the balance between user and item fairness (multi-sided fairness).

Previous studies have focused on designing algorithms that strive to balance user fairness, item fairness and overall recommendation quality (Burke et al. (2018); Wang & Joachims (2021)), yet little research exists on the effects and the tradeoffs of this multi-sided optimization. Greenwood et al. (2024) address this gap by developing a theoretical and empirical framework to analyze these effects and tradeoffs in the context of multi-sided fairness-constrained optimization. Theoretically, they concluded two phenomena: (i) the cost (i.e. decline in user fairness, defined as minimum normalized user utility) of item fairness is low when users have preferences that differ from one another, and (ii) item fairness may have a high cost on users whose preferences are misestimated, such as users who are new to the system (i.e. cold start users). They empirically measured these phenomena by creating their own recommendation system for arXiv preprints. Similar to their theoretical findings, they found the cost of item fairness to be low when users have preferences that differ from one another. However, they concluded that the cost of having misestimated users was already so high such that imposing further item fairness constraints on them did not further increase costs.

This reproducibility study focuses on examining the reproducibility of Greenwood et al. (2024) by replicating the experiments using their published code on the same arXiv dataset and Semantic Scholar data. Additionally, we extend their research in two ways. Firstly, we make an extension by examining whether the proposed claims hold for a novel dataset consisting of Amazon books reviews. This aims to evaluate the generalizability of the claims regarding multi-sided fairness tradeoffs across more widely used domains, such as book e-commerce. Recommending books is a more common application than academic paper recommendation, primarily due to its commercial relevance, which makes it a useful extension for evaluating the generalizability of the original work. Secondly, we further extend the research by tackling one of its limitations: analyzing the effects of user-item fairness tradeoffs when recommending multiple items to each user. Greenwood et al. (2024) experiment with recommending only one item, however, recommending multiple items better reflects a real-world scenario as recommendation systems typically suggest multiple items to users rather than a single one. To support this scenario, we adapted their theoretical framework and performed the same experiments.

This paper successfully reproduces the empirical findings of the experiments on the original and newly introduced dataset, further validating the claims made by Greenwood et al. (2024). Moreover, we observed a decreased cost when imposing item fairness constraints as the expected number of items recommended increases. The GitHub repository containing the code discussed in this paper can be found at the following link: `https://github.com/sanderhonig/RE-User-item-fairness-tradeoffs-in-recommendations`.

## 2 Scope of reproducibility

This research investigates the empirical claims concerning the identified phenomena by Greenwood et al. (2024). Their work focuses on the implications of including item fairness constraints alongside user fairness in a recommendation system for arXiv preprints. The fairness definition they adopt follows an *individual egalitarian* approach, meaning maximizing the utility of the worst-off individual. Therefore, their definition of user fairness is given by the normalized minimum user utility. By employing an in-processing technique for fairness that integrates item fairness directly into the recommendation algorithm, they discovered the following phenomena:

- Discovered phenomenon 1: *"When user preferences are diverse, there is 'free' item and user fairness."* Free fairness implies that item fairness can be imposed with minimal cost to the user while it is beneficial for the items.

- Discovered phenomenon 2: *"Users whose preferences are misestimated can be especially disadvantaged by item fairness constraints."*

Building on these theoretical insights, they attempted to empirically validate these phenomena using an arXiv recommendation system, leading to the following claims:

- Empirical claim 1: *"More homogeneous groups of users have steeper user-item fairness tradeoffs – as theoretically predicted, diverse user preferences decrease the price of item fairness."* Notably, the price of fairness becomes substantial only when strict item fairness constraints are imposed.

- Empirical claim 2: *"The 'price of misestimation' is high (users for whom less training data is available receive poor recommendations), but on average item fairness constraints do not increase this cost."*

This study seeks to validate these empirical claims by reproducing the experiments of Greenwood et al. (2024) on the arXiv dataset. To further explore the robustness of their claims, this paper will extend the original work by examining whether the proposed claims hold for a novel dataset about Amazon books reviews. Additionally, this research expands the original study by increasing the expected number of recommended items, allowing for a deeper analysis of the user-item tradeoff.

## 3 Methodology

For the reproduction of these empirical findings, we lean on the original theoretical framework and use the GitHub repository of the original paper [1]. We further extend their research by reproducing their claims on a novel dataset and examine the effect of increasing the expected number of recommended items per user on user fairness.

### 3.1 Theoretical framework

Greenwood et al. (2024) propose a recommendation framework that aims to balance user fairness and item fairness, leading to a multi-sided fairness problem. The principle of *egalitarian fairness* (i.e. the utility of all agents is given by the utility of the worst-off agent) is used to quantitatively define both user and item fairness, where the balance between the two is parametrized by fairness level $\gamma$. This leads to the following optimization constraint:

$$U^*(\gamma, \omega) = \max_\rho \min_i U_i(\rho, \omega)$$
$$\text{subject to} \quad \min_j I_j(\rho, \omega) \geq \gamma \max_\phi \min_j I_j(\phi, \omega) \tag{1}$$

Here, $\rho$ denotes a set of parameters referred to as the *recommendation policy* ($\phi$ is the recommendation policy for the item fairness optimization problem, $\rho$ is the recommendation policy for the user fairness optimization problem and – since subjected to the item fairness constraints – to the entire optimization problem). $\rho_i$ is the distribution over items to recommend to user $i$, with $\sum_j \rho_{ij} = 1$, $\rho_{ij} \in [0, 1]$. Utility matrix $\omega$ is calculated using the cosine similarity between user and item embeddings, where $\omega_{ij} > 0$ denotes the utility of recommending item $j$ to user $i$. $U_i(\rho, \omega)$ denotes user $i$'s *expected utility* following $\rho$ normalized by the utility they would receive from being recommended their best match, and $I_j(\rho, \omega)$ denotes item $j$'s *expected utility* following $\rho$ normalized by the utility it receives if it is recommended to every user. Their values are given by:

$$U_i(\rho, \omega) = \frac{\sum_j \rho_{ij}\, \omega_{ij}}{\max_j \omega_{ij}}$$
$$I_j(\rho, \omega) = \frac{\sum_i \rho_{ij}\, \omega_{ij}}{\sum_i \omega_{ij}} \tag{2}$$

In words, Equation 1 seeks to find the parameters $\rho$ for which the minimum normalized expected user utility is maximized (i.e. most fair according to egalitarian fairness), while the minimum normalized expected item utility is at least a fraction $\gamma$ of its optimal value. Note that for $U^*(\gamma = 0, \omega) = 1$ since the optimal recommendation policy $\rho$ would deterministically recommend each user their most preferred item.

The price of the item fairness constraint can be calculated as the normalized decrease in user fairness when subjected to the item fairness, given by the formula:

$$\pi^F_{U|I}(\gamma', \omega) = \frac{U^*(\gamma = \gamma', \omega) - U^*(\gamma = 1, \omega)}{U^*(\gamma = \gamma', \omega)} \tag{3}$$

---

[1]Accessible via https://github.com/vschiniah/ArXiv_Recommendation_Research

Utility matrix $\omega$ can be estimated based on the embeddings of users and items to be recommended, however, it should not be ignored that these utilities are mere estimations. This price of misestimation can be computed by:

$$\pi_U^M(\gamma', \omega, \hat{\omega}) = \frac{U^*(\gamma = \gamma', \omega) - \min_i U_i(\hat{\rho}(\gamma'), \omega)}{U^*(\gamma = \gamma', \omega)} \tag{4}$$

Here, $\omega$ denotes the true utility matrix, $\hat{\omega}$ denotes the estimated utility matrix, and $\hat{\rho}(\gamma')$ denotes a recommendation policy that solves the recommendation problem (Equation 1) with respect to the misestimated utilities, i.e. $\hat{\rho}(\gamma')$ solves $U^*(\gamma', \hat{\omega})$.

For a new user without known preferences (a so-called cold start user), the platform estimates their preference as the average of the preferences of existing users. This would, without item fairness constraints, result in the platform recommending generally popular items to all cold start users, regardless of their actual preference. Introducing item fairness constraints could worsen this as now not only items are recommended regardless of actual preferences, but (near) all items will be recommended including many generally unpopular ones. Therefore, theoretically one would expect user fairness to be worsened as $\gamma$ increases.

## 3.2 Datasets

In this paper, we use the arXiv dataset[2] to reproduce and extend the work of Greenwood et al. (2024). Furthermore, we use the Amazon books reviews dataset[3] to investigate how their findings generalize to a different domain, namely book e-commerce.

### 3.2.1 Original dataset: arXiv

The arXiv dataset consists of 2,639,142 papers, containing papers published on arXiv from 1991 up until now[4]. Each entry covers a paper, which is defined by fourteen attributes. Just as Greenwood et al. (2024), we only considered papers that possess one or more Computer Science categories and dropped all remaining papers. This left 707,763 papers. The train and test sets are created based on a paper's year of publication: all papers published before 2020 are selected for the train set, and all papers from 2020 are selected for the test set, resulting in 255,138 and 65,948 papers respectively. The category distributions of these sets are visualized in Appendix A.1 Figure 5. The distribution of our dataset differs slightly from the ones presented by Greenwood et al. (2024), shown in Appendix A.2 Figure 7.

Additional information on all test set papers is acquired[5] through API calls to Semantic Scholar[6], using the Semantic Scholar corpus-IDs. All papers where the corpus-ID could not be acquired are discarded, leaving 26,254 papers in the test set for further use. To match the size of the test set with the original authors, we sampled 14,307 papers such that the proportions of all subcategories remained unchanged. Section 3.3.1 explains how we again used sampling on the test set when making additional API calls to retrieve extra information for logistic regression. The category distributions of these samples are visualized in Appendix A.1 Figure 6. We kept the train set size unchanged as it was unclear how the dataset sizes by Greenwood et al. (2024) were obtained, and explicit sampling could lead to significant performance cost[7][8].

### 3.2.2 Novel dataset: Amazon books reviews

The Amazon books reviews dataset consists of 3,000,000 reviews with each review accompanied by attributes of the corresponding book, including title, author, description, and year of publication. The reviews span

---

[2]Accessible via https://www.kaggle.com/datasets/Cornell-University/arxiv

[3]Accessible via https://www.kaggle.com/datasets/mohamedbakhet/amazon-books-reviews

[4]The dataset is updated weekly. These numbers correspond to the time of writing this, January 11[th], 2025.

[5]Since these are the ones on which recommendations will take place. The train set is merely used to create embeddings for authors, as described in Section 3.3.1.

[6]More information about the Semantic Scholar API can be found via: https://www.semanticscholar.org/product/api

[7]E.g.: unlucky sampling could lead to sampling papers of which many authors are not present in the test set, making the effective size of usable authors small.

[8]Later correspondence clarified this, as discussed in Section 6.

from 1996 until the end of 2013 and include 212,404 unique books. Books that missed either a description or the year of publication were excluded from our analysis, as the description is essential for creating the user embeddings, and the year of publication is required to split the dataset. After cleaning the data, we retained 78,571 unique books. For the training set, we considered books published before 2007. This training set was used to create embeddings for the reviewers, who represent the users in the recommendation system. It contains 30,506 books, 475,382 reviews, and is contributed to by 237,310 unique reviewers. The test set included books published from 2007 until the end of 2011, and contained 14,548 books. The test set is the set of books to be recommended. We chose these years to align the test set size with the original paper.

## 3.3 Experimental setup

To replicate the original study, the README files in the paper's repository were taken as a leading reference, in coordination with the description in the paper itself. The code provided largely required minor adjustments, however, some files were missing, which we reintroduced to fully run the original pipeline. This pipeline starts with general data preparation, where the train and test set are separated and additional information about papers is gathered through API calls. Subsequently, embeddings for papers and authors, and associated similarity scores are calculated. After all preparation, logistic regression is performed to evaluate whether calculated similarity scores are a good measure of successful recommendations. Finally, we run the experiments to validate the main claims of the paper. Below we describe the detailed experimental set-up of reproducing the results on the arXiv dataset (Section 3.3.1) and of our proposed extensions: (i) validating the claims on a different dataset (Section 3.3.2), and (ii) examining the effect of user fairness when multiple items are recommended for each user (Section 3.3.3).

### 3.3.1 Original setup: arXiv dataset

We maintained the below pipeline of the original paper[9] with minor adjustments.

**General data preparation** Identically to Greenwood et al. (2024), we retrieved all papers of the arXiv dataset as described in Section 3.2.1 and discarded all papers that do not possess Computer Science as a main category. We added an extra script to produce all mappings from category-ID to main and subcategory, as this file seemed to be missing. We moved the script to gather additional features of the test set forward; according to the original README, this script is performed later, leading to problems in later scripts, hence we moved it forward. Furthermore, we introduced a script to sample the subcategories proportionally.

**Logistic regression data preparation** Additional API calls to the Semantic Scholar API are made to obtain information for logistic regression regarding: all authors that contributed to any paper in the test set, all papers that have a paper from the test set in their citation, and all citations of papers in the test set. Since we could not obtain a Semantic Scholar API key (due to a pause in issuances by high demand), we were restricted to the 1,000 API calls per second shared across all unauthenticated requests to the Semantic Scholar servers[10]. This led to many unsuccessful responses and eventually to the API server blocking our requests, as described in Appendix A.3. To mitigate this problem, we retried failed responses four more times before moving on to the next paper. Still, the number of failed responses forced us to significantly reduce the test set available for logistic regression down to $1/12^{\text{th}}$ of our initial test set by sampling proportionally to the existing subcategories, visualized in Appendix A.1 Figure 6. This ended up in successfully gathering additional information for 2,188 papers, approximately $1/6.5^{\text{th}}$ of the number of papers in the test set of Greenwood et al. (2024).

**Embeddings and similarity scores** Paper embeddings are computed for both train and test sets by considering each paper's title, abstract and categories, removing stopwords, and letting scikit-learn's TF-IDF vectorizer create embeddings. Subsequently, utility matrix ($\omega$), containing all similarity scores, is constructed in a two-step process. First, for every author (that occurs in the test set) $i$, the cosine similarity is calculated between each of their train set papers and a test set paper $j$. Say author $i$ has $n$ papers in the train set, this would result in $n$ distinct similarity scores. Second, to calculate $\omega_{ij}$, the maximum out of the $n$ similarity scores is considered. In the original code, author embeddings are created for 1,000 uniquely

---

[9]As explained in Section 6, our approach is in line with a previous version of the original paper.
[10]As stated here: https://www.semanticscholar.org/product/api

sampled authors from the test set. Since no embedding can be created for authors that do not appear in the train set, we modified the code to create embeddings for all authors present in both sets, after which we sample 1,000. The results of the original approach are shown in Appendix A.4, and the results of our approach are discussed in Section 4. Furthermore, we adjusted how the calculated similarity score is stored for logistic regression. In the original codebase, the similarity score is stored by iterating for each author over all items, starting from the first item entry every time. This effectively overwrote all previously stored similarity scores for previous authors. We corrected this by starting to write the similarity scores from the first entry of the corresponding author.

**Logistic regression** To evaluate the recommendation system, we performed logistic regression. Since the original repository did not include this code, we implemented it based on the details provided in the paper. The same sample size of 1,128 was used as the original paper. We added all recommendations per user for our logistic regression, instead of only the top-1 recommendations. The logistic regression was conducted using Python's `statsmodels` module between the similarity score and whether the author cited the recommended paper, the author was referenced, or the user was an author on the item. The similarity score was computed as described in the paragraph above. For robustness, we performed logistic regression three times with random seeds 42, 999, and 123. We calculated the coefficient, standard error, Z-value, and P-value.

**Experiments** We added a script to generate the data source file for all experiments based on the README file of the original authors, since this script was missing. For each experiment, the utility matrix $\omega$ is computed again analogously. The first experiment examines the difference in user-item fairness tradeoff between heterogeneous and homogeneous users. For heterogeneous users, we sampled 40 authors and 20 papers out of the entire test set. Then, for 50 values of $\gamma$ between 0 and 1, $U^*$ is calculated and plotted. In total 10 curves are calculated, after which the mean and (two) standard deviations are plotted. For homogeneous users, all authors are first grouped into 25 clusters. For this clustering, the original code casts the sparse into dense embeddings, resulting in significantly more memory usage. We ensured the sparse representations were maintained to significantly save time and memory without a change in result. The process of creating the remainder of the graph is identical to that of the heterogeneous graph, except that for each curve 40 authors of the same cluster are sampled. The second experiment examines the difference in user-item fairness tradeoff between users for whom preference data is present and cold start users. The latter category is constructed by treating 10% of the sampled users as cold start users by removing their embedding. Then again, for 50 values of $\gamma$ between 0 and 1, $U^*$ is calculated and plotted.

### 3.3.2 Extension: Amazon books reviews dataset

User embeddings are defined by all book embeddings to which a reviewer made a review prior to 2007. In the original paper, user embeddings were constructed using the title, abstract, and category of each item. To align with this, we generated user embeddings based on each book's title, description, and category, using scikit-learn's TF-IDF vectorizer in the same manner as the original implementation. To evaluate the recommendation system, we performed logistic regression following the methodology described in section 3.3.1, adjusted for the Amazon books reviews dataset. The logistic regression was conducted between the similarity score and whether the user actually left a review for the recommended book. The similarity score reflects the similarity between the books reviewed by the user before 2007 (i.e. user embedding), and the possible recommended books from the test set.

For the experiments, we sampled 1,000 users, consistent with the original paper. We held all experimental parameters, for conducting the experiments, constant to ensure comparability with the original paper. Specifically, we tested 50 values of $\gamma$ between 0 and 1, clustered the 1,000 users into 25 clusters using the $k$-means algorithm for the homogeneous population for the first experiment, and again treated 10% of the population to be misestimated for the second experiment.

### 3.3.3 Extension: Multiple recommendations per user

We extended the original study on the arXiv dataset by recommending more than one item to each user to better simulate a real scenario. In real-world recommendation systems (e.g. streaming, e-commerce platforms, etc.), users are often provided with multiple recommendations instead of a single item.

We noticed that the optimal policy $\rho$, which maximizes the minimal user utility (Equation 1), under the constraint $\sum_j \rho_{ij} = 1$, leads to the following without item fairness: for each user $i$, the item with index $h$ –with $\omega_{i(j=h)}$ the highest utility score for user $i$– is allocated $\rho_{i(j=h)} \to 1$, while all other items have $\rho_{i(j\neq h)} \to 0$. In other words, for user $i$ the distribution over items to recommend $\rho_i$ is a one-hot encoding. This makes sense as we aim to assign the highest recommendation weights to the items with the greatest utility to achieve the maximum expected user utility. When item fairness constraints are imposed, $\rho_i$ is *not strictly* a one-hot encoded vector anymore, but rather becomes somewhat more distributed across multiple items. The element-wise multiplication of $\rho_i$ and $\omega_i$, and subsequent summing computes the expected user utility (Equation 2). Intuitively, this can be interpreted as sampling each item $j$ with a probability $\rho_{ij}$.

When generalizing to recommending multiple ($k$) items in the optimal policy $\rho$, modifying the constraint to $\sum_j \rho_{ij} = k$, lets the $\rho_i$ distribution become a $k$-hot encoding since the constraint $\rho \in [0, 1]$ stays unchanged. This leads to the following without item fairness: for each user $i$, the set of item indices $H$ –a set of size $k$, with the $k$ highest utility scores $\omega_{i(j\in H)}$ for user $i$– is allocated $\rho_{i(j\in H)} \to 1$, while all other items have $\rho_{i(j\notin H)} \to 0$. As previously, when item fairness constraints are imposed, $\rho_i$ is *not strictly* a $k$-hot encoded vector anymore, but rather becomes slightly more distributed across multiple items. The element-wise multiplication of $\rho_i$ and $\omega_i$, and subsequent summing now leads to the expected user utility of recommending $k$ items (Equation 5). This can still be interpreted as sampling each item $j$ with a probability $\rho_{ij}$, leading to $k$ recommended items in expectation. Note that, when item fairness constraints are imposed, the top-$k$ items with the greatest utility are not necessarily chosen.

In this experiment, $k$ is set to three and five, to analyze the impact on $U^*$ of recommending a set of three, and five items respectively. As recommending multiple items automatically increases the expected user utility, the normalization of the user utility should be adjusted to maintain an equivalent range for the expected user utilities across different $k$ values. Therefore, we adjusted the normalization for user $i$'s utility in Equation 2 to:

$$U_i(\rho, \omega) = \frac{\sum_j \rho_{ij}\, \omega_{ij}}{\sum_{j\in H}\, \omega_{ij}}, \text{where } H \text{ is the set of indices with the } k \text{ highest utility scores for user } i \qquad (5)$$

This adjustment ensures that user $i$'s expected user utility is normalized by the sum of their $k$ highest utility scores. It generalizes the idea of normalizing by their highest utility score when recommending only a single item. To clarify, the only differences from the original setup in section 3.3.1 are setting $k$ to values other than 1 and modifying the normalization.

### 3.4 Computational requirements

For the experiments described in Section 3.3, we used a node containing nine cores of the Intel Xeon Platinum 8360Y, an NVIDIA A100 GPU, and 60GB of DRAM. In total, all computing time took 99 hours. We calculated the $CO_2$ emission to be approximately 12 kg $CO_2$[11].

## 4 Results

In this section, we first briefly review the original results of the paper. Secondly, we present our findings from the reproducibility study on the arXiv dataset using the original setup. Thirdly and fourthly, we discuss the results of the two extensions. Since no seed is set when sampling authors and papers for the curves, the curves vary slightly every run due to randomization.

### 4.1 Original results

Greenwood et al. (2024) validated their arXiv recommendation system by performing a logistic regression on the similarity score and whether the paper was actually cited by the author. Table 1 shows their results with a highly significant positive coefficient, confirming a reliable recommendation system.

---

[11]Taking the average Carbon Efficiency of The Netherlands from Moro & Lonza (2018) and the calculation tool of Lacoste et al. (2019).

The results of their experiments are presented in Figure 1. Figure 1a shows that item fairness imposes a higher cost on homogeneous users compared to diverse users, consistent with *empirical claim 1*. This cost remains low except when $\gamma$ reaches 1, matching strict item fairness. Figure 1b demonstrates a substantial cost of misestimation. This cost is so high that item fairness does not have a negative impact on user fairness, corresponding to *empirical claim 2*.

| | Coefficient | Standard Error | Z-value | P-value |
|---|---|---|---|---|
| Similarity score | 12.4100 | 0.058 | 212.178 | 0.000 |

Table 1: Logistic regression results to validate the recommendation system with the arXiv dataset (Greenwood et al., 2024).

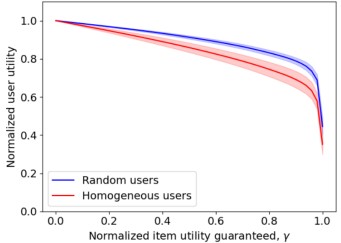

(a) Homogeneous versus diverse users on the arXiv dataset.

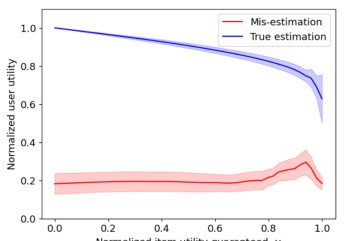

(b) With and without misestimation on the arXiv dataset.

Figure 1: The empirical findings of Greenwood et al. (2024) between user fairness (normalized minimum user utility $U^*$, Eq. 1) and item fairness constraint ($\gamma$, Eq. 1)

## 4.2 Original setup: arXiv dataset

This section presents the results of our reproducibility study on the arXiv subset. Due to API rate limits discussed in Section 3.3.1, we ran the logistic regression for the arXiv dataset on the subset of 2,188 papers instead of 14,307 papers. To ensure that using a subset of the test set for the experiments would not impact the results, we conducted the experiments on both the full test set and a subset of it. Since results on the full test set (Appendix A.5 Figure 9)) and on the subset of the test set (Figure 2) show a similar trend and no significant differences, we decided to present results on the subset of 2,188 papers for consistency with the dataset on which logistic regression is performed.

Table 2 shows the results of the logistic regression. The coefficient of 6.21 suggests a strong positive relationship between the similarity score and whether or not the author cites the paper in the future. This led us to conclude that recommending items based on similarity score is valid. The coefficient is, however, lower than what is reported in the original paper (see Table 1), which might be caused by the faulty storage of similarity scores as described in Section 3.3.1.

| | Coefficient | Standard Error | Z-value | P-value |
|---|---|---|---|---|
| Similarity score | $6.2092 \pm 0.3429$ | $0.1650 \pm 0.0093$ | $37.8554 \pm 4.0966$ | 0.000 |

Table 2: Logistic regression results using three random seeds to validate the recommendation system with the arXiv dataset.

Figure 2a demonstrates the comparison between heterogeneous and homogeneous users. The resulting curves are similar to those observed in the original paper. For moderate item fairness constraints (i.e. $0 \leq \gamma \leq 0.9$), we observe a slight negative effect on user fairness by increasing item fairness. As $\gamma$ approaches 1, the tradeoff becomes significantly steeper. Homogeneous users exhibit a higher price of fairness, consistent with *empirical claim 1* in the original study.

Figure 2b presents the results of the comparison between users with and without misestimation (i.e. users without and with prior known preferences). The curves of this experiment closely resemble those of the original paper. A similar drop in user fairness is present as for random users in Figure 2a, while user fairness stays consistent, with even a slight increase, for cold start users when increasing $\gamma$. This shows, like stated in *empirical claim 2* in the original paper, that the cost of misestimation is already so high that it does not seem to be worsened with item fairness constraints as we analyzed in Section 3.1.

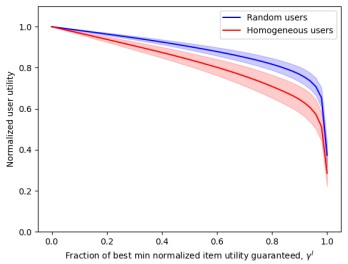

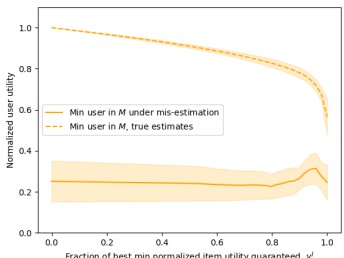

(a) Homogeneous versus diverse users on the arXiv dataset.

(b) With and without misestimation on the arXiv dataset.

Figure 2: Empirical findings between the user fairness (normalized minimum user utility $U^*$, Eq. 1) and item fairness constraint ($\gamma$, Eq. 1) by sampling from 1,000 authors from a subset of 2,188 papers.

### 4.3 Extension: Amazon books reviews dataset

This section presents the results of the original experiments conducted on the Amazon books reviews dataset. Table 3 presents the results of the logistic regression. The coefficient of 15.05 suggests a strong positive relationship between the similarity score and whether or not a reviewer leaves a review for a book in our recommendation selection. Thus, we concluded our recommendation system to be valid.

|  | Coefficient | Standard Error | Z-value | P-value |
|---|---|---|---|---|
| Similarity score | $15.05 \pm 0.78$ | $0.365 \pm 0.012$ | $41.23 \pm 1.04$ | 0.000 |

Table 3: Logistic regression results using three random seeds to validate the recommendation system with Amazon books reviews dataset.

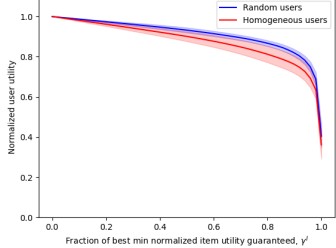

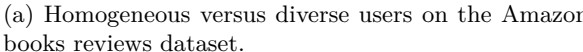

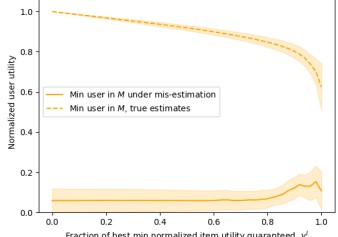

(a) Homogeneous versus diverse users on the Amazon books reviews dataset.

(b) With and without misestimation on the Amazon books reviews dataset.

Figure 3: Empirical findings between the user fairness (normalized minimum user utility) and item fairness (normalized minimum item utility), where $\gamma$ is the item fairness constraint, using the Amazon books reviews dataset.

Figure 3a demonstrates the comparison between heterogeneous and homogeneous users. The resulting curves are similar to those observed in the original paper. Homogeneous users exhibit a higher price of fairness for the novel dataset, which aligns with *empirical claim 1* reported in the original paper. Notably, the two curves are closer to each other compared to the original study. A possible explanation for this difference could be the more homogeneous nature of the Amazon books reviews dataset. Further details about this analysis can be found in Appendix A.6.

Figure 3b shows the comparison between users with and without misestimation. The curves of this experiment closely resemble those of the original paper, further supporting the robustness of the proposed *empirical claim 2* that item constraints do not increase the cost of misestimation.

## 4.4 Extension: Multiple recommendations per user

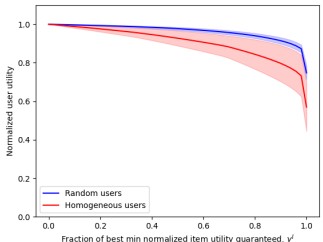

(a) Homogeneous versus diverse users for $k = 3$.

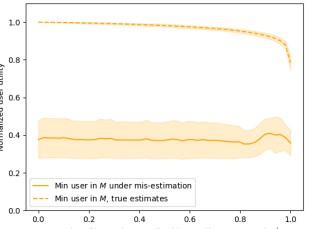

(b) With and without misestimation for $k = 3$.

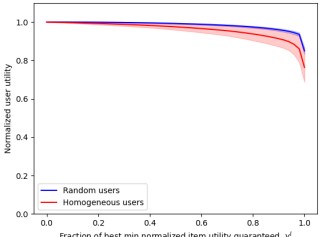

(c) Homogeneous versus diverse users for $k = 5$.

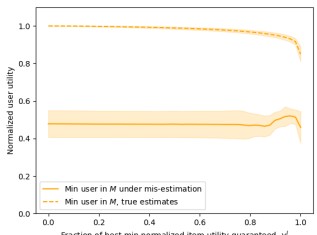

(d) With and without misestimation for $k = 5$.

Figure 4: Empirical findings between the user fairness (normalized minimum user utility $U^*$, Eq. 1) and item fairness constraint ($\gamma$, Eq. 1) by sampling from 1,000 authors from a subset of 2,188 papers. Respectively, $k = 3$ and $k = 5$ represent the recommendation of three and five items.

This section displays the experimental results of recommending multiple items, conducted on the arXiv subset of 2,188 papers. Figures 4a and 4c show the differences between homogeneous and heterogeneous users when recommending three and five items respectively. The observed curves have a similar trend to those in the original paper, where homogeneous users encounter a higher cost of item fairness than heterogeneous users, and the curves become substantially steeper when $\gamma$ approaches 1. So, this is consistent with proposed *empirical claim 1*. However, a notable observation is that both curves become less steep as $k$ increases. Intuitively, this makes sense as the negative impact of an individual recommendation is mitigated by considering a larger variety of items. Due to this enlarged variety in recommendations, the gap between recommendation differences for homogeneous and heterogeneous users decreases.

Figures 4b and 4d illustrate the disparity between misestimated and truly estimated users when recommending three and five items. As expected, the line of misestimated users shifts upwards for a higher $k$ since misestimations are less problematic when more items are recommended. In this context, when recommending more arXiv preprints, the likelihood of the user encountering a relevant paper increases, which leads to a higher (normalized minimum user) utility. Furthermore, we observe that item constraints do not increase the cost of misestimation, which aligns with *empirical claim 2* in the original paper.

# 5    Discussion and conclusion

We reproduced the empirical framework described in the original paper by Greenwood et al. (2024). First, we performed logistic regression on the subset of the original dataset, confirming that similarity scores provide a valid estimation for future citations which act as a proxy for user preferences. We then performed the same empirical experiments on both the original and novel Amazon books review dataset to prove generalizability. Secondly, we extended the experiments to examine the effect of increasing the number of recommended items per user on user fairness with item fairness constraints.

The results of both datasets validate *empirical claim 1*: (i) diverse user preferences decrease the price of item fairness and (ii) the price of item fairness becomes substantial when $\gamma$ approaches 1. We similarly validate *empirical claim 2*: the price of misestimation is high, but imposing item fairness constraints does not increase this cost. Additionally, increasing the expected number of recommended items does not influence these two claims, however, the cost of imposing item fairness does seem to lessen.

The revalidation of *empirical claim 1* further validates the findings of the original paper. These observations are in line with points discussed in earlier research on item fairness in recommendation systems. Zhao et al. (2025), Koutsopoulos & Halkidi (2018), and Castellini et al. (2023) investigated the relationship between user fairness, item fairness, and diversity in recommendation systems, and similarly observed that increasing user diversity leads to improved item fairness. Zhao et al. (2025) attributed this to the fact that, in this manner, generally unpopular items have naturally a higher chance of being recommended. This corresponds to our observation of a decreased price of item fairness when user diversity is higher. The importance of this diversity is further elaborated on by Kunaver & Požrl (2017), however, it is considered beyond the scope to discuss it in this reproducibility study.

The revalidation of *empirical claim 2* again shows evidence that the theoretical intuition of further worsening user fairness when increasing item fairness for cold start users does not seem to hold, instead user fairness is stagnantly low with a slight increase near strict item fairness (observed by us and the original paper). We suspect that, due to some of the naturally present item variability in recommendations for cold start users, it is of no value to impose minimum variability up to a certain level (by item fairness). When increasing minimum variability beyond this level, while recommendations are still not personalized, the increased diversity raises the odds of providing better matches. Since (normalized) *minimum* utility measures the most underserved user in the cold start group, near strict fairness uplifts the bottom – the one user who had terrible matches before now has a higher probability to get something relevant, nudging the minimum up. However, as the graph drops at strict item fairness, there seems to be an optimum; strict levels will also force the recommendations of the most unpopular items to some users, dropping user fairness again. It would be valuable to further investigate this hypothesis in future work. Future work should investigate the relation between the amount of variability in item recommendations and (normalized minimum) user utility for cold start user. Additionally, we suggest running the experiments more than 10 times to obtain statistically significant results for a better-founded conclusion.

Increasing the number of recommended items $k$ from one to three and five showed no influence on these two claims. However, increasing $k$ does seem to lessen the cost of imposing item fairness. The recommendation of one, three or five papers, made up a substantial part of the entire recommendation pool of 20 papers. Future work could investigate whether this effect weakens on a larger sample size, i.e. when a smaller portion of the total items is recommended. This would further decrease the high standard deviation we observed due to our small sampling. [12]

The revalidation of both claims on the Amazon books review dataset demonstrated the generalizability beyond the domain of academic paper recommendations to a more widely used domain. The item-user fairness curves between heterogeneous and homogeneous groups were slightly closer to each other than in the original work, likely caused by the more homogeneous nature of users within the dataset. Future research could explore datasets with more heterogeneous random users, as well as those from other widely adopted

---

[12]Larger sampling was already implemented in a newer version of the original paper, but due to time and computational limitations, we were unable to do this ourselves, as discussed in Section 6.

domains (e.g. social media, music streaming, etc.). Since recommendation systems are broadly applied, enhancing robustness across different domains is a valuable direction for future research.

While we have addressed the limitation of recommending only a single item, various other constraints of the original work still remain. These include the need to theoretically explore the user-item tradeoffs to other definitions of fairness, the challenge of platform hesitation to adopt an egalitarian approach, and the limited number of user types in the theoretical proofs (in Appendices D and E of the original paper). Addressing these limitations is still a relevant area for future research, and could enhance the framework's applicability to real-world scenarios.

Overall, in line with the original findings, our results suggest that recommendation system designers should keep in mind that it is beneficial to have a diverse user population and that item fairness constraints should be imposed on the entire population rather than subgroups.

## 6 Final remarks

**How we deviated from the original authors.** Due to time constraints, limited computing resources and the computational complexity of optimizing the convex problem, we were confined to reproduce the older original NeurIPS version of the paper, accessible via `https://neurips.cc/virtual/2024/poster/94638`. This version deviates from the most recent version in three aspects. Firstly, it draws smaller samples of papers (20 instead of 200) and authors (40 instead of 500) for each curve. Secondly, it first samples 1,000 papers to sample each curve from. Both the old and new versions of the paper share very similar graphs and identical conclusions. Thirdly, for creating homogeneous authors, it creates more clusters (25 instead of 10). These choices resulted in significantly less memory usage and faster computing times, with very similar plots and an identical conclusion section. Lastly, as described in Section 3.3.1, we ensured the clustering for creating homogeneous groups was performed on sparse embeddings.

**What was easy.** The original code was easy to find and publicly available. Together with a well-organized repository, it allowed us to reproduce their results. Furthermore, the original research already extensively discussed several extensions in their appendix, answering quite some of our questions and therefore enabling efficient progress. The foundation of enabling multiple-item recommendations was already present in the original code, enabling easier integration of this extension.

**What was difficult.** Loading the original dataset from Kaggle resulted in more data than described by the original authors, which made it unclear whether the originally reported number of papers in the train and test set was obtained. Also, some adjustments in code and execution order were necessary to obtain a working pipeline, which was challenging since only parts of the code contained comments. Moreover, to obtain additional information for each paper for logistic regression, we needed data from the Semantic Scholar API. Lacking an API key, we were unable to make all the requests needed to get the data for every paper. Besides, the original paper did not provide code or a clear explanation to perform the logistic regression, hence we had to make assumptions solely based on the paper. Furthermore, extending their theoretical framework for multiple-item recommendation was challenging, because their framework did not explicitly incorporate that only one item is recommended, only implicitly in both their framework and code.

**Communication with original authors.** During this replication study, we contacted the authors for the following two points: Since the linked GitHub repository did not provide a file for the logistic regression, we asked for clarification on their implementation. In their answer, they provided the code for the logistic regression, which led us to conclude that our implementation slightly differed. Greenwood et al. (2024) ran logistic regression three times: depending on whether a citation was present, whether the author was referenced, and whether the user was an author on the paper. The first metric was discussed in their paper. Our implementation assumed that if either of those were true, it was a good recommendation, and used this as a variable for the logistic regression. Secondly, our train and test set sizes differed significantly from their train and test set sizes. They clarified that they have used the 'update date', while we made use of the 'published date'. Since many papers have been updated recently, Greenwood et al. (2024) have significantly fewer papers in their train and test sets. Due to time constraints, we were not able to change both of these deviations.

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

# A   Appendix

## A.1   arXiv dataset distribution of this study

This section provides the details regarding the categories distribution of the arXiv dataset used in this reproducibility study. Figure 5 shows all papers published *before* and *in* 2020 in the arXiv dataset at the time of writing. Out of all the papers published in 2020, we sampled 14,307 papers to align with the test set size of Greenwood et al. (2024), the distribution of which is shown in Figure 6a. Due to the failed API calls required for logistic regression –as described in Section 3.3.1– our final test set contained 2,188 papers published in 2020, of which the distribution is shown in Figure 6b.

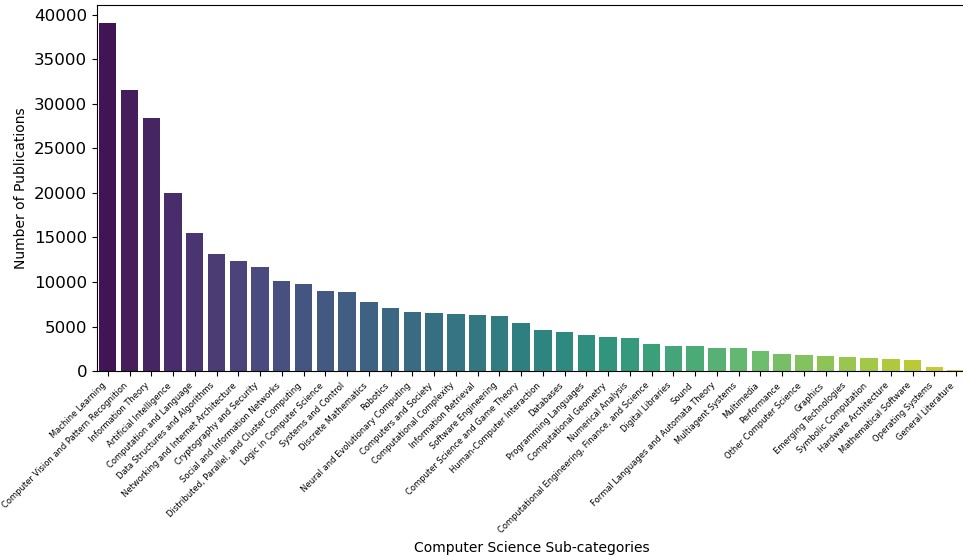

(a) Distribution of papers by subcategory of the 255,138 papers in the train set (i.e. papers published before 2020).

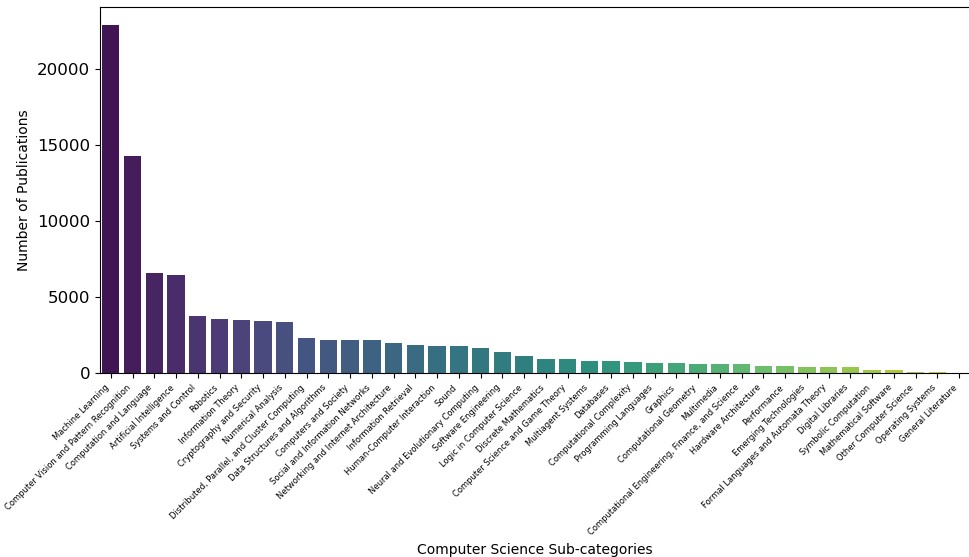

(b) Distribution of papers by subcategory of the 65,948 papers published in 2020.

Figure 5: Distribution of papers by subcategory per dataset. A paper possessing multiple subcategories is counted multiple times.

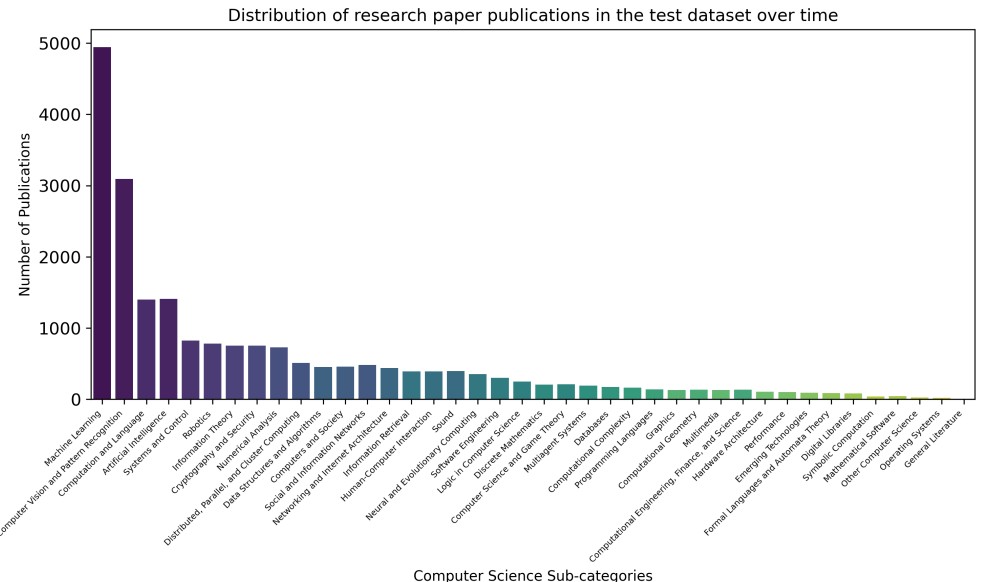

(a) Distribution of papers by subcategory of the 14,307 sampled test set papers. These papers were used for experiments.

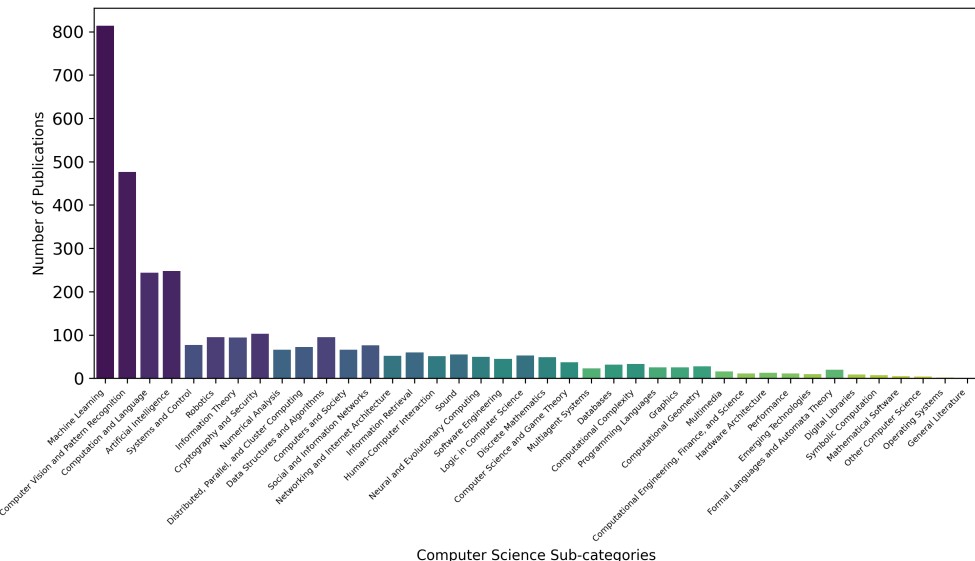

(b) Distribution of papers by subcategory of the 2,188 sampled test set papers for which Semantic Scholar API calls succeeded. These papers were used for experiments and for logistic regression.

Figure 6: Distribution of papers by subcategory per sampled dataset. The x-axis order of categories is equivalent to Figure 5b for comparability. A paper possessing multiple subcategories is counted multiple times.

## A.2   arXiv dataset distribution of original paper

Figure 7 provides the details regarding the categories distribution of the arXiv dataset used by Greenwood et al. (2024).

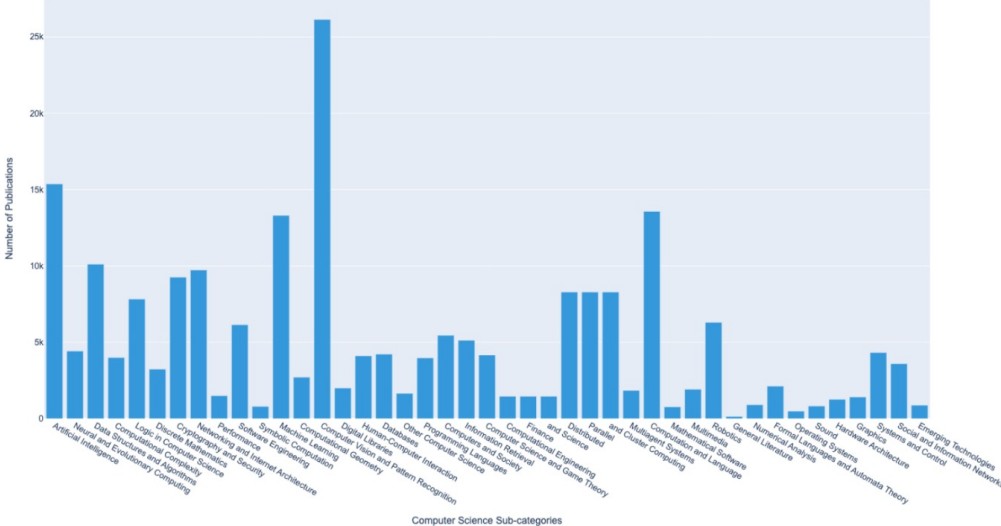

(a) Distribution of papers by subcategory in the train set (i.e. papers published before 2020).

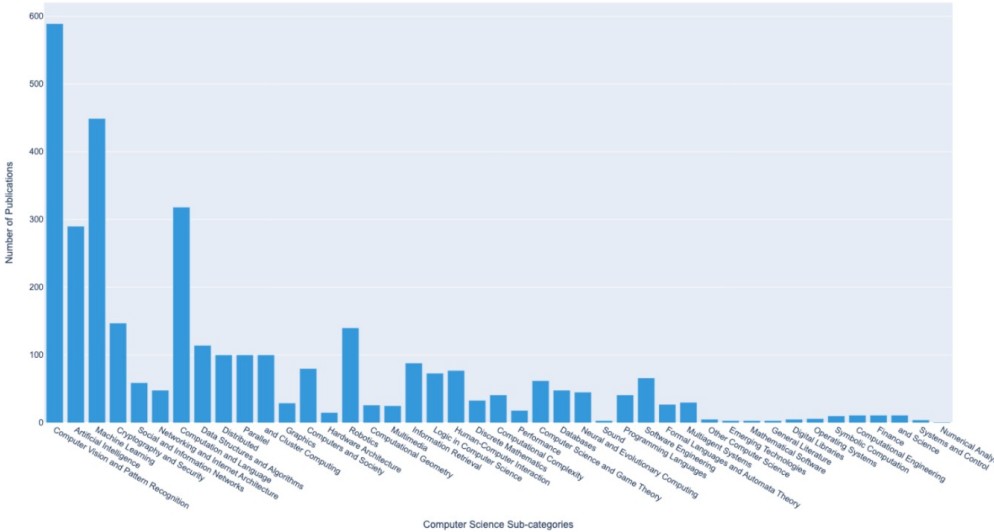

(b) Distribution of papers by subcategory in the test set (i.e. papers published in 2020).

Figure 7: Distribution of papers by subcategory per dataset (Greenwood et al., 2024).

### A.3 API rate limit error

When making an API request when more than 1,000 other unauthenticated requests were performed, the following error was given:

```
429:  Too Many Requests.  Please wait and try again or apply for a key for a higher rate
limits.
```

After a large number of requests (sometimes in the hundreds, sometimes in the few thousands), occasionally the server kicked all requests for dozens of minutes and the Python script was terminated. This is likely caused by the server blocking our IP as a common rate-limiting strategy.

```
requests.exceptions.ConnectionError:  HTTPSConnectionPool(host='api.semanticscholar.org'
, port=443):  Max retries exceeded with url: *URL* (Caused by NameResolutionError
("<urllib3.connection.HTTPSConnection object at 0x7fd8db3a7f20>:  Failed to resolve
 'api.semanticscholar.org' ([Errno -3] Temporary failure in name resolution)"))
```

### A.4 Test set sampling method in provided code

Figure 8 shows plots produced by the sampling method in the originally provided code. This method differed from what was described in the paper, by how authors were sampled to generate the curves. The original code first sampled 1,000 authors randomly from the test set, after which embeddings were made for authors that also appeared in the training set. This resulted in a significantly smaller pool of authors (659 authors) from which 40 authors were sampled to generate each curve in the plots. Contrary, the results in the body of the paper first created all author embeddings, after which 1,000 authors were sampled.

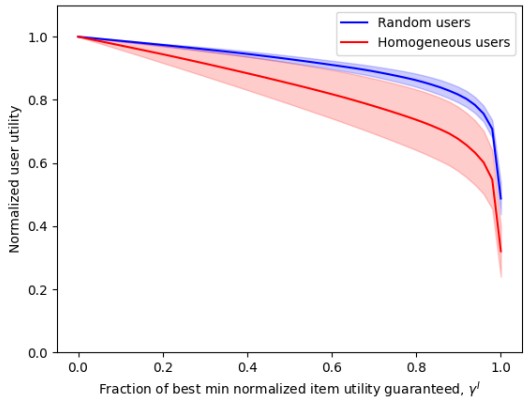
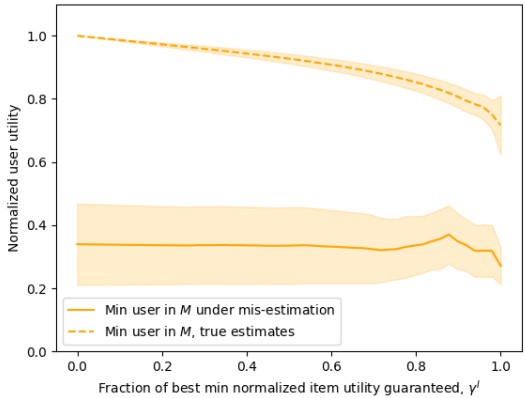

(a) Homogeneous versus diverse users on the original dataset.

(b) With and without misestimation on the original dataset.

Figure 8: Empirical findings between the user fairness (normalized minimum user utility $U^*$, Eq. 1) and item fairness constraint ($\gamma$, Eq. 1) by sampling from 659 authors from a subset of 2,188 papers.

### A.5 Test set sample of 14,307

Figure 9 shows the empirical results of the full test set. The smaller sample in the paper aligns with the dataset on which logistic regression was performed. These two sets differed due to the difficulties with API calls, as explained in Section 3.3.1.

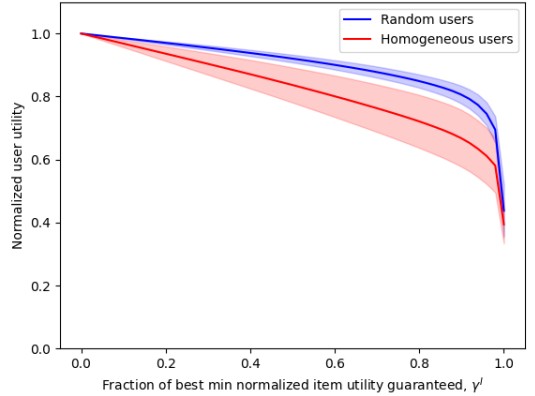
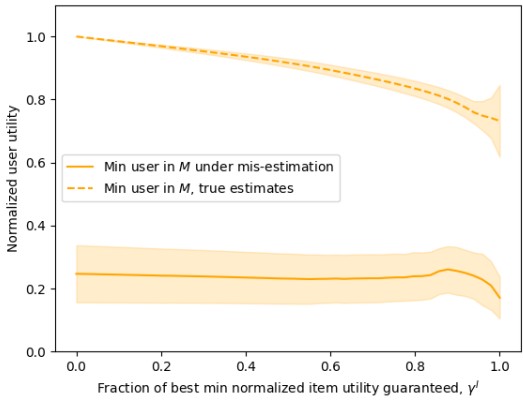

(a) Homogeneous versus diverse users on the arXiv dataset.

(b) With and without misestimation on the arXiv dataset.

Figure 9: Empirical findings between the user fairness (normalized minimum user utility $U^*$, Eq. 1) and item fairness constraint ($\gamma$, Eq. 1) by sampling from 1,000 authors from a subset of 14,307 papers.

### A.6   Amazon books reviews homogeneity

For the Amazon books reviews dataset, homogeneous users seemed less negatively impacted by item fairness than suggested by the original arXiv dataset. A possible explanation for this difference is that homogeneous users in the Amazon books reviews dataset may be clustered more heterogeneously. To test this, we computed the silhouette scores for clusters in both datasets, a metric measuring within-cluster similarity relative to other clusters, with lower scores indicating greater heterogeneity. However, as Table 4 shows, both datasets yield similar scores, albeit hinting towards a slightly more heterogeneous distribution of the arXiv dataset.

|  | Silhouette score |
|---|---|
| arXiv preprints dataset | $0.3401 \pm 0.0038$ |
| Amazon books reviews dataset | $0.3447 \pm 0.0096$ |

Table 4: Silhouette scores for three random seeds (42, 999, 123).

To further investigate why the two curves are closer to each other, we plotted the authors and the clusters to which they belong. Figure 10 shows a greater homogeneity of authors in the Amazon books reviews dataset; the user embeddings are gathered in a small interval of values, compared to the arXiv dataset with more spread. This leads to a higher probability of more similar users being randomly sampled as heterogeneous users in the Amazon dataset compared to the arXiv dataset. The more homogeneous nature of the random users in the Amazon dataset can lead to more similar user utility between the two curves. This can explain the two curves being closer to each other compared to the original study on the arXiv dataset.

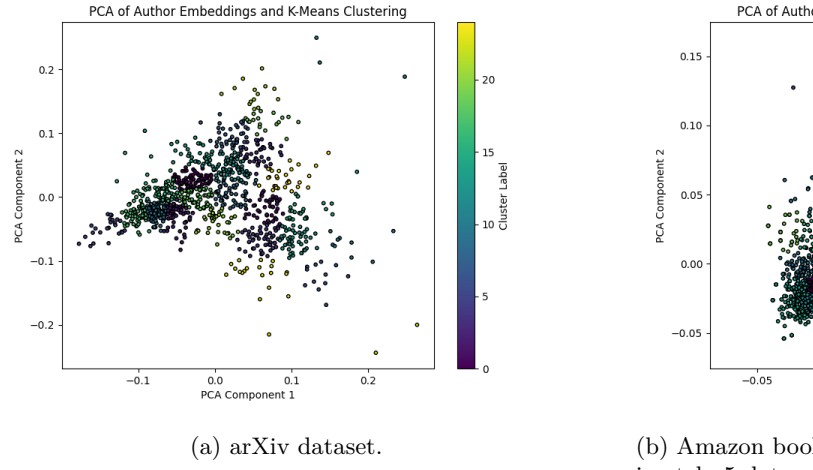

(a) arXiv dataset.

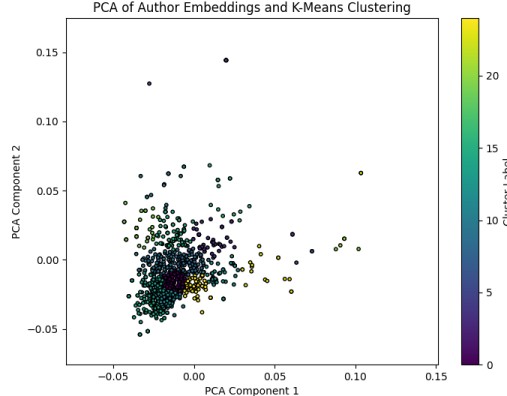

(b) Amazon books reviews dataset. We left out approximately 5 data points that were outliers.

Figure 10: Scatterplots for the test set of both datasets depicting 25 clusters and two PCA components.

### A.7 Amazon books reviews dataset

This section provides details regarding the data distribution of the Amazon books reviews dataset. Figure 11 and Figure 12 show the distribution of books and reviews, respectively, based on their publication date.

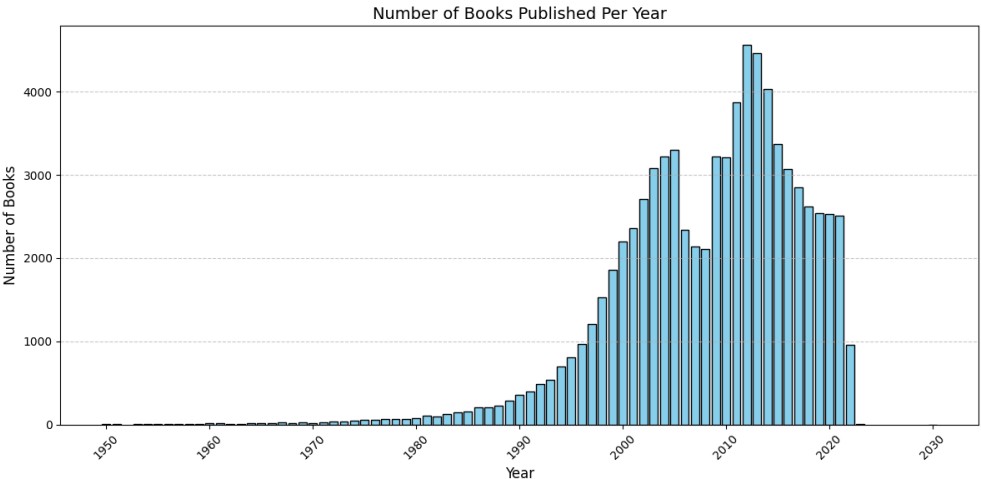

Figure 11: Number of books published per year in the Amazon books reviews dataset (48 books between 1802 and 1950 were dropped for visualization).

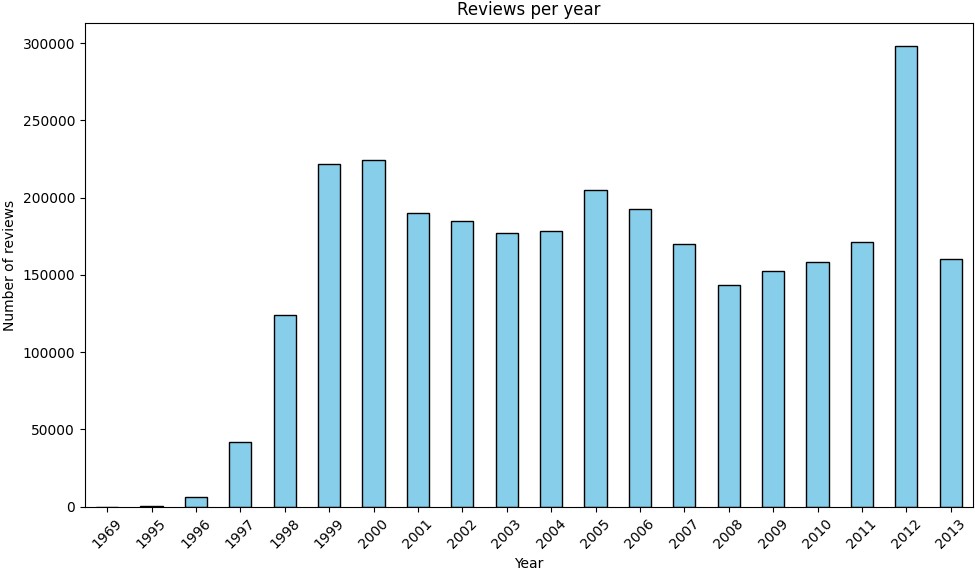

Figure 12: Number of reviews per year in the Amazon books reviews dataset.

