# OpenReview forum: "A reproducibility study of “User-item fairness tradeoffs in recommendations”"
_TMLR — Accepted by TMLR_

### Review · Reviewer_Bv8X · 2025-03-23

**Summary Of Contributions:**

The paper reproduces the empirical analysis in [[Greenwoord et al., 2024]](https://arxiv.org/abs/2412.04466) on the trade-off between recommendations quality and fairness, and extends it to both a new dataset and to the case of multiple recommended items for each user. Their empirical results strengthen the empirical findings already proposed in the original paper.

**Audience:**

Yes

**Claims And Evidence:**

Yes

**Requested Changes:**

**Q1:** The paper is not extremely clear when it is exposing the entire pipeline for the arXiv dataset experiment. Given that this is the core result of the work (as it is the reproduction of the original results), I would suggest to make the text clearer, especially on the parts describing embedding preparation, their scoring and the following logistic regression process.

**Q2:** The most interesting aspect of [[Greenwood et al., 2024]](https://arxiv.org/abs/2412.04466) to me is the discrepancy between their theoretical claim that fairness has an high cost for misestimated users, while the empirical results suggest that such a cost mainly stems from the misestimation itself, and fairness is instead even alleviating the problem somehow. I would have liked to see some more discussion on this, as a valuable contribution could be to better understand this gap, and try to give the respective merits to the fairness constraint and to the misestimation part. How can we measure this aspect more precisely? Is the impact of fairness really negligible when users are misestimated?

**Q3:** The work is absolutely not related to any existing literature other than its reference [[Greenwood et al., 2024]](https://arxiv.org/abs/2412.04466)... This does not help really is positioning the current contribution within the current landscape, and indeed pointing out how your work is similar or differ from other similar analyses (for example those I mentioned above) would greatly help is making your own contribution stand out from the work.

**Strengths And Weaknesses:**

Assessing the impact of AI technologies that we commonly see deployed in the wild, such as recommender systems, is always an interesting research direction. Given their ubiquitous presence in almost every online platform or service, their (possibly unintended) effects are a critical aspect that regulators and policy makers need to take into account. However, the paper provides a poor advancement in our understanding of these aspects: the main focus in on reproducing a setting that has already been proposed and investigated by another existing work (with little modifications to their whole pipeline), basically only corroborating their findings. The second, novel, investigated dataset adds little to these findings, given that all the results go (as expected indeed) in the same direction. The most interesting part is that extending the results to multiple recommendations for each user, which is indeed not present in the original work used as a basis. However, also these results are a bit expected, and also similar works, such as [[Castellini et al., 2023]](https://papers.ssrn.com/sol3/papers.cfm?abstract_id=4428125), [[Fletcher et al., 2023]](https://papers.ssrn.com/sol3/papers.cfm?abstract_id=4319311) and [[Calvano et al., 2023]](https://papers.ssrn.com/sol3/papers.cfm?abstract_id=4448010), already proposed and investigated similar issues and properties. I am unsure about how the current work is adding value to the already existing literature in its current form.

---

> ### Author Response · Authors · 2025-04-16
>
> Thank you for taking the time to provide us with your thoughtful and structured feedback. We hear your concern regarding the novelty of insights in our paper. Our vision and aim of the paper --in light of being a reproducibility study-- is to add value by investigating the reproducibility, investigating the generalizability, and giving a wider perspective on this research.
>
> Your feedback gave us insights into the shortcomings of especially the last point. Concretely, we made the following changes based on your feedback:
>
> **Q1:** Thank you for pointing out the unclarity of the pipeline of for the arXiv dataset experiment. We now better structured and more thoroughly elaborated in Section 3.3.1 on the three aspects you mentioned: embedding preparation, similarity scoring, and the logistic regression process.
>
> **Q2:** Greenwood et al.'s finding was indeed interesting and remarkable. As our focus for this reproducibility study was to reproduce and discuss their findings, we did not include this extra discussion initially. However, upon your remark, we agree with the added value of providing a more thorough reflection on this empirical finding with our intuition and potential further ideas for further research. Therefore, we added a paragraph describing exactly this in the *Discussion and conclusion* section.
>
> **Q3:** We agree with your opinion about the value of including further existing literature to better position the paper within the current landscape of research. We restructured the Discussion and conclusion section and now added a paragraph describing the relation of our findings to earlier research regarding empirical claim 1 in the Discussion and conclusion section, including the insights of Castellini et al. (2023) and additional references. Fletcher et al. (2023) and Calvano et al. (2023) specifically focus on market behavior. Since their research did not relate to user-item fairness tradeoffs and did not provide conclusions, we chose not to include these references. For empirical claim 2, we now include a more thorough elaboration on our findings, as touched on in our answer for *Q2*.

---

### Review · Reviewer_XJP2 · 2025-03-30

**Summary Of Contributions:**

This paper focuses on replicating and verifying the empirical findings of the proposed method by Greenwood et al. (2024) and the key contributions including successfully replicating the main results and extending the experiments with a new Amazon Books Review dataset to further validate the original paper's results.

**Audience:**

No

**Claims And Evidence:**

Yes

**Requested Changes:**

Please refer to the Weaknesses part.

**Strengths And Weaknesses:**

Strengths:
- The authors successfully replicated the experiments from Greenwood et al. (2024), although they encountered challenges due to time constraints.
- The paper is well-structured and the reasoning is presented clearly.

Major Weaknesses:
- Lack of novel insights beyond the confirmation of the original findings (for example, more generalizable insights regarding failure modes, unexpected behaviors, and limitations of the original work). While validating the empirical results of the original paper is important, identifying its limitations and drawbacks is also valuable.

- Only a single new dataset (Amazon Books Reviews) is explored, and the random users in this dataset are inherently more homogeneous. This weakens the conclusion on robustness and is also linked to the weakness mentioned above, which fails to provide more generalizable insights. The original paper uses a narrow domain to empirically validate their framework, which is not representative of other recommendation settings like e-commerce or social media. Recommendation systems in practice have very broad application scenarios. This new dataset’s structure and task are very similar to the arXiv one. Why not try other widely adopted benchmark datasets in the area of recommendation systems, such as MovieLens, Last.fm.....?

- In the original paper, Greenwood et al. mentioned that the framework is restricted to a small number of user types and assumes symmetric utilities. While the authors extend this to alternative fairness definitions (section 3.3.3) and multiple recommendation per user (section 4.4), the scope still remains narrow. A broader exploration of user types and utility models should be integrated to enhance the generalizability of the findings and better align with real-world application scenarios.

---

> ### Author Response · Authors · 2025-04-16
>
> Thank you, we really appreciated the feedback you provided us with. Based on your suggestions, we incorporated the following changes:
>
> - Change 1: Initially, we didn't consider it necessary to restate the limitations of the original paper, as they were already stated in that paper. However, we understand your concerns about highlighting them for a complete reproducibility analysis. Therefore, we added these limitations in the discussion and conclusion.
> - Change 2: We consider the Amazon books review dataset, being part of the e-commerce domain, more widely used than paper recommendation, making it a valuable dataset to evaluate the generalizability of the original work in other domains. We have clarified this argumentation in the introduction. While we agree that it is not ideal that the dataset is inherently more homogeneous, we doubt whether the alternative datasets align more closely to the arXiv dataset. It would indeed enhance robustness to extrapolate this method to even more datasets, but unfortunately, that was not feasible due to time constraints. Therefore, we now explicitly mention this as a direction for future research in the discussion and conclusion.
> - Change 3: In this research, we focused on replicating the original work on the arXiv and Amazon books review dataset. Additionally, we addressed one important limitation of the original framework regarding multiple-item recommendation. Unfortunately, the other limitations remain outside the scope of our study. However, we do mention these limitations now (see change 1). Furthermore, we don’t consider the symmetric utility model a big limitation, since they argue in Appendix A2 that it often captures the basic structure behind producer preferences.

---

### Review · Reviewer_egGN · 2025-04-03

**Summary Of Contributions:**

This paper is a replication study of the paper "User-item fairness tradeoffs in recommendation" by Greenwood et al. (2024). This paper reproduces the empirical results of Greenwood et al. (2024) which showed in a recommender system for arXiv preprints that 1) the tradeoff between recommendation user fairness and item fairness is lessened in populations of users with more heterogeneous preferences, and 2) the cost to user utility of mis-estimating user preferences is high but not increased by imposing item fairness constraints. The paper also validates these claims by extending the empirical analysis of Greenwood et al. to a different dataset of Amazon book reviews, and shows that the same results hold. Finally, the authors note that Greenwood et al. only analyzed the setting of providing a *single* recommended item, and extend the empirical analysis to recommendation sets with $k > 1$ recommended items, observing similar patterns but a flatter user-item fairness tradeoff.

**Audience:**

Yes

**Claims And Evidence:**

Yes

**Requested Changes:**

## Critical changes

1. It would be helpful if the authors could provide further implementation details on the Amazon book recommender.
2. I propose that the authors should either (a) justify why the generalized Equation 1 will produce one/$k$-hot policies for all values of $\gamma$, or (b) reframe this section to clarify that the recommendation policy is not produce a recommendation set but rather a distribution from which a set (of expected size $k$) can be sampled, and argue that this will resemble the tradeoffs that would be seen in top-$k$ recommendation.

## Additional minor changes (not critical)

1. At the bottom of page 1 the authors state that "the cost (i.e. decline in user utility) of item fairness is low" and that Greenwood et al. "found the cost of item fairness to be low for users with diverse preferences". In the context of Greenwood et al., the cost is the decline in the user *fairness* (that is, minimum user utility), and the cost of item fairness is low for *populations of users* with diverse preferences, that is, when users do not agree with each other on which items are "good" and "bad" (as opposed to individual users' preferred items being a diverse set of items). I would recommend the authors clarify the phrasing to reduce ambiguity.
2. Section 3.1 states that Greenwood et al. "propose a recommendation system that aims to balance user fairness, item fairness, and overall recommendation quality"; however, Greenwood et al. do not propose a recommendation *system* but propose a *framework* for studying the fairness tradeoff, and this framework does not take into account the overall recommendation quality.
3. What is the distribution of subcategories in the test set papers the authors were able to obtain from the Semantic Scholar API, and how does this differ from the distribution of the authors' full test set?
4. The authors state that "in the original codebase, the similarity score is stored by overwriting all previously stored similarity scores for each author. We corrected this by writing the similarity score to the data entries which belong to the author it relates to". I'm not sure I understand the issue and amendment here -- it would be helpful if the authors could provide further explanation.

**Strengths And Weaknesses:**

## Strengths
- The authors provide a comprehensive description of their process of reproducing Greenwood et al. (2024), including clear descriptions and justifications for necessary changes.
- The authors provide results on a new, distinct dataset from a different recommendation setting, which shows not only that the original analysis is replicable but also that the original results replicate _beyond_ the original setting.
- The paper explores several hypotheses to explain the observation that the tradeoff curves for homogeneous users and for randomly sampled users are closer together in the new Amazon dataset compared with the arXiv dataset.

## Weaknesses
- The paper does not provide details for how the embeddings were generated for the new Amazon dataset.
- In Section 3.3.3, which discusses the generalization to top-$k$ recommendations, the authors claim that the recommendation policy which solves Equation 1 (optimizes user fairness subject to item fairness) converges to a one-hot vector, and use this claim to imply that replacing the constraint $\sum_j \rho_{ij} = 1$ with $\sum_j \rho_{ij} = k$ results in a $k$-hot recommendation policy, so that this new optimization problem finds a recommendation set of $k$ items. While this is true for when there are no item fairness constraints, the theoretical results of Greenwood et al. include a closed-form solution to Equation 1 when item constraints are maximal, and this solution is not necessarily one-hot -- rather, this recommendation policy defines a distribution from which a single item is sampled. I suspect that using the constraint $\sum_j \rho_{ij} = k$ will not produce a $k$-hot policy. This policy could be used to construct a recommendation set by including item $j$ in the set with probability $\rho_{ij}$, but this set would have *expected* size $k$ rather than deterministically have size $k$.

---

> ### Author Response · Authors · 2025-04-16
>
> Thank you very much for your valuable feedback. In the following part, we will clarify the changes we made based on your feedback.
>
> Critical changes:
> 1. The embeddings of the Amazon dataset were created in a similar manner to that of the original dataset. We see how this might not have been completely clear. So we added some text to make it clearer.
> 2. Thank you very much for your detailed observation regarding the theoretic formulation. It is indeed the case that the recommendation policy represents rather a distribution than a deterministic vector. We observed this as well during our experiments, but we agree with you that it is not clearly stated in the paper. To address this, we adjusted section 3.3.3.
>
> Additional minor changes (not critical):
> 1. Our original phrasing was indeed a bit ambiguous. We revised the text to make it clearer.
> 2. Indeed! We changed "system" to the "framework" in section 3.1.
> 3. We agree that the exclusion of this information might raize questions. Thank you for raising this oversight; we now included, and referred to, graphs in the Appendix showing the distribution of subcategories for both samples of the test set we used in our experiments.
> 4. Thank you for bringing this unclarity to our attention. We rephrased this now.

---

> > ### Comment · Reviewer_egGN · 2025-04-30
> >
> > Thank you for your response!
> >
> > For the histogram of test set subcategories, it's hard to tell to what extent the distribution changed between the two test sets-- a reader needs to examine how the x-axis labels are rearranged. I wondering if making the histogram x-axis ordered by subcategory (but potentially keeping the ranking color-coding) would make this easier to parse.
> >
> > In the rephrased Section 3.3.3., you wrote "In this setting, the $k$ items with the highest utility tend to receive the highest recommendation weights. As a result, we expect the top-$k$ recommendations to align with this sampled set of expected size $k$." I'm not sure what this argument is saying; is this referring to the case _without_ item fairness constraints (in which case, it is the case that the solution will certainly be a k-hot vector for the $k$ items with the highest utility?). If you are also referring to the case _with_ item fairness constraints, it would be good to justify this further.
> >
> > Aside from these small points, your changes address all my concerns.

---

> > > ### Author Response · Authors · 2025-06-05
> > >
> > > Thanks again for your feedback. We incorporated the following changes based on your remarks:
> > > - We adjusted the histograms of the test set subcategories in Figures 6a and 6b such that the x-axis order of categories is equivalent to Figure 5b for comparability.
> > > -  In Section 3.3.3, we clarified the following sentences further: "In this setting, the k items with the highest utility tend to receive the highest recommendation weights. As a result, we expect the top-k recommendations to align with this sampled set of expected size k."

---

### Decision · Action_Editor_jEL1 · 2025-05-09

**Recommendation:** Accept with minor revision

**Comment:**

Two of the reviewers recommend acceptance while one recommends rejection. Overall, I do not see the reviewers' concerns as serious in the context of a reproducibility study and thus recommend acceptance with minor revision. The revision should address the following remaining comments from Reviewer egGN:
1. In Section 3.3.3, please clarify the sentences "In this setting, the $k$ items with the highest utility tend to receive the highest recommendation weights. As a result, we expect the top-$k$ recommendations to align with this sampled set of expected size $k$." Their meaning is also not clear to me.
1. Please make the histograms of test set subcategories (Figures 6(a) and 6(b) I believe) more comparable as recommended by the reviewer.
1. The last line on page 1 still has the ambiguous phrase "users with diverse preferences."

**Audience:**

For a reproducibility study, the criterion for sufficient interest is whether the study provides generalizable insights (see [TMLR acceptance criteria](https://jmlr.org/tmlr/acceptance-criteria.html)). In my view, this criterion is satisfied because the study generalizes the original findings to a second Amazon book dataset and to the setting of multiple recommended items. I agree with the authors that Amazon book recommendation is a more widely applicable use case than arXiv paper recommendation in the original work.
- While I acknowledge Reviewer XJP2's desire for further generalization to more distant use cases and to more complicated settings, I think the current level of generalization is sufficient. The authors also added more discussion on these points for future work.
- Reviewer Bv8X commented that the findings on the Amazon dataset and for multiple recommendations were as expected, which I also do not see as a problem, especially given that this is a reproducibility study. In response to Reviewer Bv8X, the authors related empirical finding 1 to additional prior works, which improves "interest".

**Claims And Evidence:**

This submission is a reproducibility study of "User-item fairness tradeoffs in recommendations" by Greenwood et al. (2024). The study empirically validates the two main claims of the original paper:
1) the cost of imposing item fairness is lower for more heterogeneous populations of users,
2) item fairness does not further increase the cost for users whose preferences are mis-estimated.

The study also generalizes the original work in two directions:
1) replicating the main findings on a second dataset of Amazon book reviews,
2) generalizing to recommendation of multiple items as opposed to a single item in Greenwood et al.

Reviewers praised the clarity of description and reasoning for the authors' process of reproduction and did not find major issues. There were questions about how embeddings were computed for the Amazon dataset and about the overall pipeline, which the authors addressed during the rebuttal period.